# Genetic Diversity Analysis and Breeding of Geese Based on the Mitochondrial ND6 Gene

**DOI:** 10.3390/genes14081605

**Published:** 2023-08-10

**Authors:** Yang Zhang, Shangzong Qi, Linyu Liu, Qiang Bao, Teng Wu, Wei Liu, Yu Zhang, Wenming Zhao, Qi Xu, Guohong Chen

**Affiliations:** Key Laboratory for Evaluation and Utilization of Poultry Genetic Resources of Ministry of Agriculture and Rural Affairs, Yangzhou University, Yangzhou 225009, China; zyang@yzu.edu.cn (Y.Z.); qsz8200@163.com (S.Q.); liulinyu9812@163.com (L.L.); dx120200144@stu.yzu.edu.cn (Q.B.); tengwu2022@163.com (T.W.); mz120211462@stu.yzu.edu.cn (W.L.); yuzhang@yzu.edu.cn (Y.Z.); wmzhao@yzu.edu.cn (W.Z.); xuqi@yzu.edu.cn (Q.X.)

**Keywords:** local breeds, mitochondrial DNA, differences in body shape, genetic diversity, dynamic analysis

## Abstract

To explore the differences in body-weight traits of five goose breeds and analyze their genetic diversity and historical dynamics, we collected body-weight data statistics and used Sanger sequencing to determine the mitochondrial DNA of 100 samples of five typical goose breeds in China and abroad. The results indicated that Lion-Head, Hortobagy, and Yangzhou geese have great breeding potential for body weight. Thirteen polymorphic sites were detected in the corrected 505 bp sequence of the mitochondrial DNA (mtDNA) ND6 gene, accounting for approximately 2.57% of the total number of sites. The guanine-cytosine (GC) content (51.7%) of the whole sequence was higher than the adenine-thymine (AT) content (48.3%), showing a certain GC base preference. There were 11 haplotypes among the five breeds, including one shared haplotype. We analyzed the differences in the distribution of base mismatches among the five breeds and conducted Tajima’s D and Fu’s Fs neutral tests on the historical dynamics of the populations. The distribution of the mismatch difference presented an unsmooth single peak and the Tajima’s D value of the neutral test was negative (D < 0) and reached a significant level, which proves that the population of the three species had expanded; the Lion-Head goose population tends to be stable. The genetic diversity of Lion-Head, Zhedong White, Yangzhou, and Taihu geese was equal to the average diversity of Chinese goose breeds. The Hortobagy goose is a foreign breed with differences in mating line breeding and hybrid advantage utilization.

## 1. Introduction

China has numerous high-quality domestic waterfowl breeds, including 31 swan genetic resources in the China Livestock and Poultry Genetic Resources Poultry Chronicle, 26 local goose breeds, and 1 cultivated breed. This high-quality genetic resource provides a solid foundation for the sustainable and balanced growth of the goose industry [1,2]. Genetic diversity is an important component of biodiversity. Genetic diversity is the basis for the formation of biodiversity and guarantees the evolutionary potential of species. China has very rich genetic resources for livestock and poultry and one of the largest geese populations in the world [3]. However, with the increasingly shrinking living environments of waterfowl and the lack of conservation facilities and systems [4], the gene pools of some goose breeds have not been supplemented or perfected.

Mitochondrial DNA (mtDNA) is considered to be the best molecular genetic marker to study the origin, evolution, and classification of species because of its maternal inheritance, rich base polymorphisms, and simple structure [5]. mtDNA genes have been widely used as biomolecular diagnostic markers and molecular genetic markers in the field of livestock [6,7,8]. Among the NADH(nicotinamide adenine dinucleotide hydride) dehydrogenases, the NADH dehydrogenase subunit 6 (*ND6*) gene has a relatively short sequence length, a simple structure without introns, a moderate evolution rate, abundant sequence variations, and a significantly higher mutation rate than cell nuclear DNA and has been widely used in the population genetic structure analysis of domestic cattle [9], domestic fowl [10], and insects [11].

The Lion-Head goose is the only large goose species in China, possessing the advantages of fast growth and resistance to rough feeding. Studies indicate that this breed is an ideal model to explore bird growth and development. The fat deposition in the liver and leg muscles of the Lion-Head goose is significantly higher than that of other goose species [12]. The Taihu goose has high fecundity and early maturity; it is usually used as a hybrid female parent to improve the egg production performance of hybrid offspring [13]. The Zhedong White goose is a medium-sized goose with both meat and a fatty liver. Ren et al. [14] amplified mitochondrial genes and noted that these genes could be used as the basis for the identification of some goose breeds. The Hortobagy goose is native to Hungary and is a well-known medium-sized goose for both eggs and down [15]. The Yangzhou goose was China’s first locally bred medium-sized goose breed that produced both eggs and meat [16,17]. Previous studies have mostly focused on the mitochondrial *D-loop* region and the *COI* gene. For these five large, medium, and small goose breeds from home and abroad, research on the genetic diversity and evolution of mitochondrial *ND6* has not been reported.

In this study, three domestic large, medium, and small geese and two foreign medium-sized goose breeds, the Zhedong White goose and Hortobagy goose, were used as samples. These breeds were studied through statistics of their body weight at different ages and the determination of mtDNA *ND6* gene sequences with genetic diversity and historical dynamic analyses of goose breeds was used to provide a theoretical basis for the conservation, hybrid selection, development, and utilization of local goose breeds in China.

## 2. Materials and Methods

### 2.1. Ethics Approval

All animal experiments were approved by the Institutional Animal Care and Use Committee of the Yangzhou University (approval number: 132-2022). All procedures were performed strictly in accordance with the Regulations for the Administration of Affairs Concerning Experimental Animals (Yangzhou University, China, 2012) and the Standards for the Administration of Experimental Practices (Jiangsu, China, 2008).

### 2.2. Animals and Sample Collection

According to the pedigree records, 100 blood samples were collected from unrelated geese (Table 1; Figure 1). Additionally, we selected four goose breeds: Lion-Head (LH; n = 20), Zhedong White (ZD; n = 20), Hortobagy (HB; n = 20), and Taihu (TH; n = 20). The geese were provided by the National Waterfowl Resource Bank (Taizhou, China) for the weight measurements and analysis. Approximately 20 healthy male Yangzhou geese were raised at the Yangzhou Tiange Goose Industry Co., Ltd. (Yangzhou, China), according to the standard practices of the farm (Yangzhou (YZ) geese; n = 20). We divided the 100 geese into 5 groups by variety, with 20 in each group (5 ♂ + 15 ♀/Group). Then, we collected 5 sets of eggs produced and weighed them for hatching. All individuals in each group who participated in the experiment had blood taken.

Blood samples (approximately 3 mL) were collected from the wing veins of the geese in individual vacuum blood collection tubes. Genomic DNA was extracted using a genomic DNA extraction kit (TIANGEN, Beijing, China). RNA was purified to remove genomic DNA (gDNA), miRNA, and rRNA using an RNeasy Micro kit (Qiagen, Hilden, Germany), an RNase-Free Dnase kit (Qiagen), and a RiboZero^TM^ MagneticKit (Epicentre, Illumina, San Diego, CA, USA), respectively. Data collection and analyses were conducted at the College of Animal Science and Technology, Yangzhou University, Jiangsu Province, China.

### 2.3. Determination of Knob Morphological Index

The 505 bp (goose) segment of the mtDNA control region was PCR-amplified using the primers 5′-GTTTTTTTTAGGGATTTGCT-3′ (F-Primer) and 5′-AACCGCCCGAATAGCAC-3′ (R-Primer), which were designed according to the goose sequence (accession number NC_012920.1) in GenBank (Figure 2). The PCR assay was conducted in a 20 μL reaction mixture containing 2 μL of a buffer (10 ×), 2.5 μL MgCl2 (25 mM), 1.0 μL dNTPs (10 mM), 1 μL of each primer (10 μM), 0.2 μL of Taq DNA polymerase (5 U/μL) (Takara Biomedical Technology (Beijing) Co., Ltd.), 1.0 μL of a DNA template (100 ng), and 11.3 μL sterilized water. The reaction was brought to a final volume of 20 μL by adding double-distilled autoclaved water. PCR amplification was performed using an Eppendorf Master Cycler (Eppendorf, Hamburg, Germany). The thermocycle was set as 95 °C (5 min), 94 °C (45 s), 58 °C (45 s), and 72 °C (45 s) for 35 cycles and a final elongation step at 72 °C for 5 min. The last elongation step was extended to 8 min at 72 °C and samples were held at 4 °C. The F-Primer and R-Primer were provided by Qingke Bioengineering Co., Ltd. (Nanjing, China) for the direct bidirectional sequencing of target fragments within the PCR products using the ABI 3730 XL sequencing platform.

### 2.4. Statistical Analysis

The weight difference data recorded during the experiment were analyzed using IBM SPSS 25.0 software for the analysis of variance [18]. Sequences of the mtDNA fragments were visualized using Chromas2.4.1 software and manually browsed to ensure the accuracy of the bases. The amplified mtDNA *ND6* sequence was edited and corrected using MEGA 11 software [19]. DNA SP 6.0 software was used for the AMOVA (analysis of molecular variance) analysis of the sequences of different goose breeds and to perform a statistical analysis of parsimony-informative sites, conserved sites, haplotype numbers, nucleotide differences, haplotype diversity, and nucleic acid diversity [20,21]. The haplotype ratio was calculated as the ratio of the number of haplotypes to the sample size. The results of the final neutrality tests were analyzed using Arlequin 3.1 software and GraphPad Prism 8 was used to draw the base mismatch difference analysis graph [22]. Finally, the median-joining (MJ) network of the control region of the mtDNA haplotypes was constructed using Popart v.1.7 software [23].

## 3. Results

### 3.1. Analysis of the Difference in Body Weight at Different Ages

The weight of Lion-Head geese was significantly higher than that of the other four breeds at the egg stage and ages of 1, 7, 14, 35, and 70 days (*p* < 0.05) (Table 2). At 1 d of age, the body weights of Zhedong White and Yangzhou geese were significantly higher than those of Hortobagy and Taihu geese (*p* < 0.05). The body weight of Yangzhou geese was significantly higher than that of Hortobagy, Zhedong White, and Taihu geese at 7 d of age (*p* < 0.05). The body weight of Yangzhou geese was significantly higher than that of Hortobagy, Zhedong White, and Taihu geese at 14 d of age (*p* < 0.05). At the age of 35 days, the Lion-Head, Hortobagy, and Yangzhou geese were significantly heavier than the Zhedong White and Taihu geese (*p* < 0.05). At the age of 70 days, the body weights of Yangzhou, Hortobagy, and Zhedong White geese were significantly higher than those of Taihu geese (*p* < 0.05).

### 3.2. Genomic DNA Extraction and PCR Amplification

Genomic DNA was extracted from the blood of five geese and used as a template to amplify the *ND6* fragment of mtDNA. Electrophoresis detected a clear band consistent with the size of the expected target fragment (Figure 3), which was used for the next sequencing step.

### 3.3. Genetic Diversity

#### 3.3.1. Analysis of *ND6* Locus Information and Nucleic Acid Diversity in Goose Mitochondrial DNA

Table 3 presents the locus information and nucleotide diversity of the goose mitochondrial DNA *ND6*. As shown in Table 3, using DNA SP 6 software, the corrected 505 bp mtDNA *ND6* sequences of the five varieties were homologously compared and 13 polymorphic sites were found, accounting for approximately 2.57% of the total points (12/505). The G+C content (51.7%) of the sequence was slightly higher than the A+T content (48.3%), showing a certain GC base preference, which was consistent with the characteristics of the bird mitochondrial DNA base composition. Eleven haplotypes were detected, of which Holdobagy geese had only one haplotype and no polymorphic sites. The polymorphic loci of the Zhedong White geese, Yangzhou geese, and Taihu geese were greater than three and the number of haplotypes ranged between two and three. The haplotype diversity was greater than 0.1 (Hd > 0.1), the acid diversity was greater than 0.001 (Pi > 0.001), and the nucleotide difference between Zhedong White geese and Taihu geese was greater than 0.5 (K > 0.5).

#### 3.3.2. Nucleotide-by-Nucleotide Comparison of Goose Haplotype Genes

The distribution of the unique haplotype variable sites of the mtDNA *ND6* gene in the five goose breeds is shown in Figure 4. Haplotypes 1 and 3 had the most variable sites, with four sites each. The variable sites of haplotypes 4 and 5 contained at least one variable. The number of geese with haplotype 1 was the largest and there were only one or two geese with the other haplotypes. As all insertions and deletions were removed from the analysis, the variable types were classified as transitions and transversions.

#### 3.3.3. Haplotype Distribution and Frequency Analysis of mtDNA *ND6* Gene

The distribution and frequency of unique mtDNA *ND6* haplotypes in the five goose breeds are shown in Table 4. The Lion-Head geese had the largest number of unique haplotypes, with three haplotypes accounting for 27.27% of their frequency. The Zhedong White goose, Yangzhou goose, and Taihu goose each with two haplotypes accounted for 18.18% of the frequency and the Holdobagy goose accounted for 9.09%. Among them, haplotype 1 was shared by five breeds of geese and haplotype 2 was shared by Yangzhou geese and Zhedong White geese (Figure 5).

### 3.4. Historical Dynamic Analysis

Based on the mitochondrial *ND6* gene sequence, we used DNA SP 6 software to conduct a neutrality test analysis and a base mismatch analysis of the Lion-Head goose, Zhedong White goose, Yangzhou goose, and Taihu goose populations. The results showed that the neutral test-Tajima’s D value of Taihu, Yangzhou, and Zhedong White geese were negative and the Fu’s Fs value was positive, all of which were 0.10 > *p* > 0.05 (Table 5). Combined with the base difference mismatch distribution map, the expected value of the gene pairwise difference in the three population control regions was a smooth curve, while the actual value had a single peak. This indicated that the mismatch distribution of the mitochondrial *ND6* gene sequence base difference in Taihu, Yangzhou, and Zhedong White geese was unimodal (Figure 6), which was consistent with the unimodal curve mode of population expansion and the neutrality detection results. Both Tajima’s D and Fu’s Fs values of the Lion-Head goose neutral test were negative and the difference was not significant (*p* > 0.10) (Table 5), indicating that the Lion-Head goose population was in line with the neutral mutation results and that the Lion-Head goose population was in a stable state. Combined with the base difference mismatch distribution map, it was found that the expected and actual values of the gene pairwise difference in the population control area were both smooth curves, indicating that the mismatch distribution of the Lion-Head goose mitochondrial *ND6* gene sequence base difference distribution was multimodal (Figure 6). This did not conform with the unimodal curve pattern of population expansion, which was consistent with the neutral detection results.

## 4. Discussion

### 4.1. Analysis of Body-Weight Difference of Five Breeds of Geese at Different Ages

Body-weight gain is an important indicator of animal growth and development [24,25]. In the goose production process, hybridization is usually used to make full use of heterosis to improve production indicators such as body-weight gain in offspring. Wang et al. [26] showed that the average body weights of 10-week-old Langde and Sichuan White geese were significantly higher those that of Sichuan White geese and heterosis was obvious. Huang et al. [27] found that the average feed/weight ratio of the hybrid offspring of Xingguo Grey geese and Yangzhou geese was better than that of their parents. The five geese breeds used in this study included domestic large, medium, and small geese and foreign medium-sized geese. Lion-Head geese are the only local large goose breed in China. At 1, 7, 14, and 70 d of age, the body weight was significantly higher than that of the other four breeds; however, the reproductive performance was low and the average annual egg production was only 28 [28]. Taihu geese have high egg-production rates and good prospects for supporting the production of meat. They are usually used as a hybrid female parent to improve the egg production performance of hybrid offspring [29,30]; however, their small size is a disadvantage. At the age of 35 days, the body weights of Lion-Head, Holdobagy, and Yangzhou geese were significantly higher than those of Zhedong White and Taihu geese. Among them, the weight gain of Holdobagy geese from day 1 to day 70 occurred, indicating developmental potential and weight breeding value.

### 4.2. Genetic Diversity Analysis of Five Breeds of Geese

Measuring differences in the species genetic structure depends on indicators such as the average nucleotide difference (K), nucleotide diversity (Pi), and haplotype diversity (Hd). In general, the higher the haplotype diversity (Hd) and nucleotide diversity (Pi) values, the greater the genetic variation. The population with the greater genetic variation generally has a greater selection potential [31]. Species with a high genetic diversity produce offspring that are more adaptable to the range in which they live [32]. In the current study, based on the mitochondrial DNA *ND6* sequence, five parameters-including the number of polymorphic sites, the number of haplotypes, the GC content, the haplotype diversity index, and the nucleotide diversity index were used to study the genetic diversity of goose breeds. Partial sequences of the *ND6* genes of the five geese breeds were obtained by PCR amplification and the average GC content (51.7%) was slightly higher than that of AT (48.3%), indicating a more obvious base bias. This was because the nucleotides of the protein-coding gene that cause mutations were under less natural selection pressure at the codon site. This was consistent with the results reported by Grzegorczyk et al. [33]. Haplotype (Hd) and nucleotide (Pi) diversity are important indicators of the genetic diversity of a population [34]. Li et al. determined the mtDNA *D-loop* sequences of 26 goose breeds and six Lande geese, among which the average Hd of the Chinese goose breeds was 0.1384 and the Pi was 0.00029 [35]. In the current study, the overall distribution patterns of the Lion-Head, Zhedong White, Yangzhou, and Taihu geese populations showed a haplotype diversity of Hd ≥ 0.154, a nucleotide diversity of K ≥ 0.308, and a nucleotide diversity Pi ≥ 0.00069, indicating that although the genetic diversity of the four goose breeds was higher than the average level of Chinese goose breeds, the overall nucleotide diversity remained low. This may have been because the different samples selected for the experiment led to differences in the gene sequences. Among the five breeds of geese, except for the Holdobagy goose, the proportion of unique haplotypes was high and the proportion of shared haplotypes was low (h = 1), indicating that the four domestic goose breeds were less affected by foreign breeds and that the species were conserved. The distribution pattern of the Holdobagy goose population showed low haplotype diversity (Hd = 0) and low nucleotide diversity (Pi = 0.000); hence, it may be necessary to improve the genes of local goose breeds to improve production performance by carrying out high-intensity and wide-ranging artificial selection and breeding. This would reduce the genetic diversity of the population. These findings were consistent with the experimental results of Bihui et al. [36]. Therefore, there must be suitable living environmental conditions [37] and artificial selection in the future breeding of varieties.

### 4.3. Historical Dynamic Analysis of Five Breeds of Geese

Analyses of population phylogenetic trees are increasingly used to infer the process of the historical evolution of populations. Population history evolution is usually detected using two methods: a population-based neutrality test such as Tajima’s D and Fu’s Fs analyses, or a base difference analysis. In the current study, the neutrality test (Tajima’s D) values of Taihu, Yangzhou, and Zhedong White geese were negative (D < 0) and the distribution of base difference mismatches showed a Poisson distribution [38], indicating that Taihu, Yangzhou, and Zhedong White geese may have experienced a certain degree of population expansion over a long period. However, Tajima’s D value and Fu’s Fs value of the Lion-Head goose neutral test were both negative and the difference was not significant (*p* > 0.10), indicating that the Lion-Head goose population was in line with the neutral mutation result and that the Lion-Head goose population was in a stable state. A unimodal curve pattern that did not fit the population expansion was consistent with the neutral detection results. This indicated that the population was not undergoing species expansion or continuous growth [39,40] and that large-scale migration may have occurred in the early stage. In future studies, artificial selection should be increased.

## 5. Conclusions

Lion-Head, Zhedong White, and Yangzhou geese had greater potential for body-weight selection in this study. The Holdobagy goose is a foreign breed that is genetically distant from the other four goose breeds. The population genetic diversity of Lion-Head, Zhedong White, Yangzhou, and Taihu geese was higher than the average level of Chinese geese and was relatively stable. Poultry breeding utilizes variations within and between populations to improve traits of interest. The upcoming era will promote the diversification of varieties and different characteristics within varieties and may shift from high-input to low-input production systems. Therefore, there is an urgent need to plan and protect the genetic resources of Chinese goose breeds and develop appropriate planning strategies to protect them against existing genetic variations.

## Figures and Tables

**Figure 1 genes-14-01605-f001:**
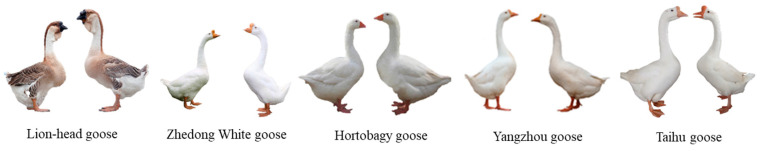
Images of goose population. Images were captured using a digital camera (Olympus).

**Figure 2 genes-14-01605-f002:**
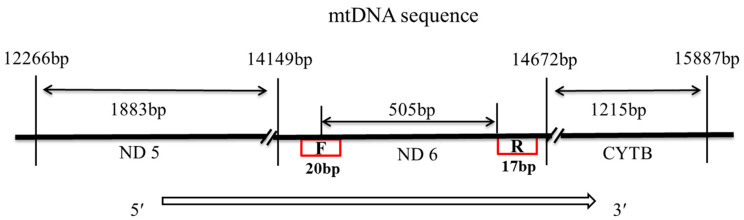
The site of primers according to the goose sequence (GenBank accession number NC_012920.1).

**Figure 3 genes-14-01605-f003:**
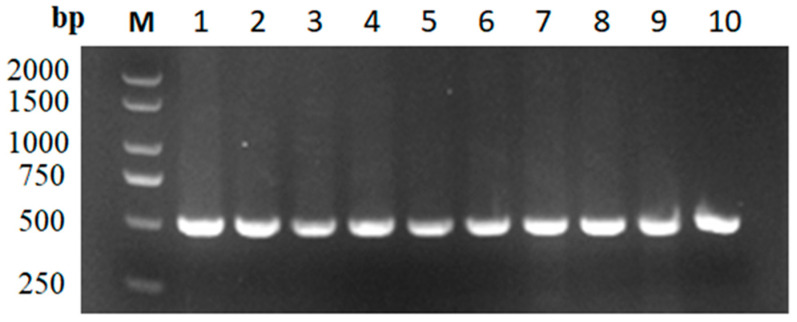
The partial electrophoresis results of PCR amplification for 2 mtDNA fragments (Olympus, Japan). Note: M: DL1500 marker; 1–10: PCR products of *ND6*.

**Figure 4 genes-14-01605-f004:**
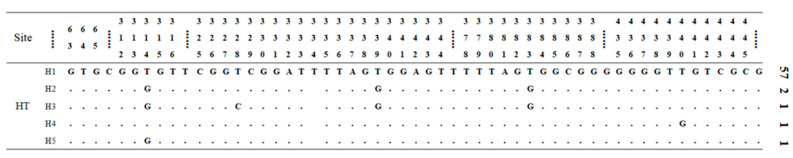
Variable sites in the mtDNA control region for haplotypes from five breeds of geese. “·” indicate identical nucleotides. Number on far-right indicates the number of individual geese with the haplotype. HT: haplotype type.

**Figure 5 genes-14-01605-f005:**
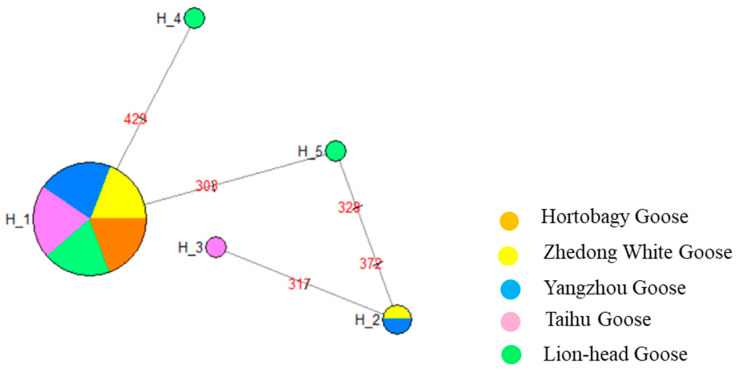
The median-joining networks of mtDNA *ND6* haplotypes. The colors represent different geese populations.

**Figure 6 genes-14-01605-f006:**
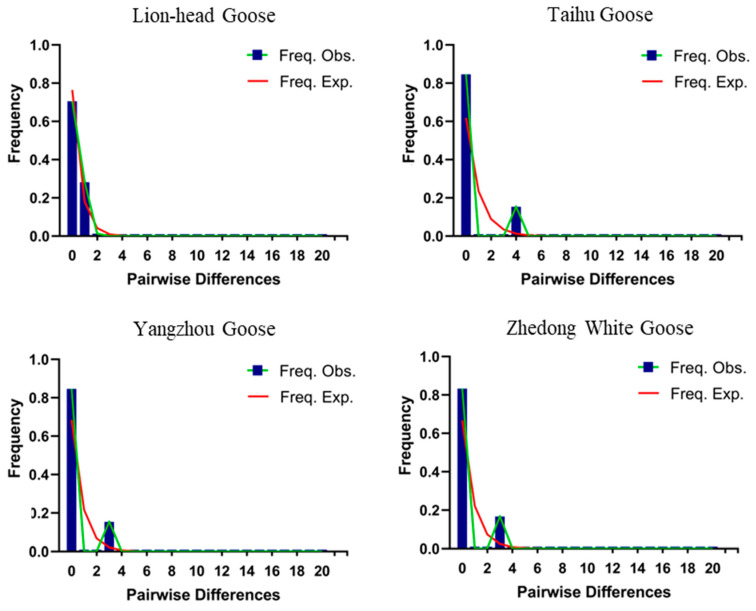
Distribution of base differential mismatches of goose mitochondrial *ND6* gene sequences. Freq. Obs. represents the actual value; Freq. Exp. represents the expected value.

**Table 1 genes-14-01605-t001:** Sample information.

Breed (Abbrev.)	Sample Size	Main Physical Features	Sample Source
LH	20	Brown plumage; tall body; high slaughter rate	Guangdong Province, China
ZD	20	White feathers; strong adaptability; tender meat	Zhejiang Province, China
HB	20	Strong disease resistance; high cashmere content	Hungary, Europe
YZ	20	High feed-conversion rate; delicious meat; no nesting	Jiangsu Province, China
TH	20	Strong disease resistance; high fecundity	Jiangsu Province, China

LH: Lion-head goose; ZD: Zhedong White goose; HB: Holdobagy goose; YZ: Yangzhou goose; TH: Taihu goose.

**Table 2 genes-14-01605-t002:** Analysis of the difference in body weights of five species of geese at different days of age.

Weight/g	LH	ZD	HB	YZ	TH
Egg weight	197.1 ± 11.19 ^a^	165.3 ± 11.38 ^b^	148.7 ± 7.53 ^c^	169.96 ± 6.45 ^b^	127.5 ± 6.37 ^d^
1 day old	128.4 ± 15.72 ^a^	107.4 ± 7.05 ^b^	89.5 ± 1.56 ^c^	111.3 ± 10.62 ^b^	83.1 ± 8.00 ^c^
7 days old	245.1 ± 17.25 ^a^	177.0 ± 8.53 ^c^	174.2 ± 6.02 ^c^	197.1 ± 20.21 ^b^	168.1 ± 5.72 ^c^
14 days old	543.6 ± 37.31 ^a^	454.5 ± 17.14 ^c^	472.5 ± 45.67 ^c^	512.9 ± 28.01 ^b^	307.6 ± 18.95 ^d^
35 days old	1939.4 ± 147.68 ^a^	1509.5 ± 11.52 ^b^	1923.0 ± 57.44 ^a^	1870.6 ± 107.60 ^a^	1180.6 ± 30.59 ^c^
70 days old	5220.1 ± 208.66 ^a^	4125.4 ± 124.66 ^c^	4362.9 ± 219.24 ^b^	4257.9 ± 256.57 ^b,c^	3157.2 ± 160.97 ^d^

Note: LH: Lion-head goose; ZD: Zhedong White goose; HB: Holdobagy goose; YZ: Yangzhou goose; TH: Taihu goose. There is no significant difference between the same letters (*p* > 0.05), and the difference between different letters is significant (*p* < 0.05).

**Table 3 genes-14-01605-t003:** Goose mitochondrial DNA *ND6* locus information, haplotypes, and nucleotide diversity.

Breed	NP	M	GC	h	Pi	Hd	K
LH	2	444	52.1%	3	0.308	0.295	0.00069
ZD	4	447	51.7%	3	0.500	0.318	0.00148
HB	0	434	51.6%	1	0.000	0.000	0.00000
YZ	3	432	51.1%	2	0.462	0.154	0.00106
TH	4	437	51.8%	2	0.615	0.154	0.00140

LH: Lion-head goose; ZD: Zhedong White goose; HB: Holdobagy goose; YZ: Yangzhou goose; TH: Taihu goose; NP: non-parsimony-informative loci; M: monomorphic sites; GC: guanine-cytosine content; h: number of haplotypes; Pi: nucleotide diversity; Hd: haplotype diversity; K: average number of nucleotide differences.

**Table 4 genes-14-01605-t004:** Number and frequency of haplotypes of five species of geese.

Breed	Unique Haplotype	Frequency
LH	3 (Hap 1, 4, 5)	27.27%
ZD	2 (Hap 1, 2)	18.18%
HB	1 (Hap 1)	9.09%
YZ	2 (Hap 1, 2)	18.18%
TH	2 (Hap 1, 3)	18.18%

There was a shared haplotype (Hap 1), with a frequency of 9.09%. LH: Lion-head goose; ZD: Zhedong White goose; HB: Holdobagy goose; YZ: Yangzhou goose; TH: Taihu goose.

**Table 5 genes-14-01605-t005:** Analysis of the historical dynamics of four types of goose population.

Item	LH	TH	YZ	ZD
Fu’s Fs	−1.4010	1.4740	0.976	1.0540
*p*-Value	*p* > 0.10	0.10 > *p* > 0.05	0.10 > *p* > 0.05	0.10 > *p* > 0.05
Tajima’s D	−1.46801	−1.77497	−1.65231	−1.62929
*p*-Value	*p* > 0.10	0.10 > *p* > 0.05	0.10 > *p* > 0.05	0.10 > *p* > 0.05

LH: Lion-head goose; ZD: Zhedong White goose; HB: Holdobagy goose; YZ: Yangzhou goose; TH: Taihu goose.

## Data Availability

All data generated or analyzed during this study are included in this published paper.

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
