# Peer review of "Genetic Diversity Analysis and Breeding of Geese Based on the Mitochondrial ND6 Gene"

_genes, 2023, doi:10.3390/genes14081605_

Round 1
Reviewer 1 Report
The manuscript reports the results of a genetic comparison of popular meat breeds of Chinese geese.
Introduction: nicotinamide adenine dinucleotide hydride
nuclear DNA
It is unclear how many breeds were studied: five domestic and two foreign in the Introduction but five breeds in 2.2 Animals and Sample Collection.
sex of geese - all male?
At the age of 35 days,... were significantly heavier....
page 9 rephrase to combine two sentences - In the current study, ... parameters including the.... index were selected... breeds.
page 10 remove . before Zhedong
minor corrections
Author Response
请参阅附件。

Reviewer 2 Report
How can you be sure that the animals were not related?
Check if the name in figure 1 is correct.
in the statistical analysis they mention the word AMOVE, they should define properly what they are referring to. I guess they are talking about analysis of molecular variance (AMOVA). The quotations related to this topic (18 and 19) are not the most appropriate to refer to this type of analysis.
They do not mention in material and methods how nucleotide diversity was calculated and with which methods and software the haplotype nucleotide diversity; diversity and the Average number of nucleotide differences.
Figure 4 is not easy to read as the authors put it in the paper.
The first part of the conclusions should be more direct, it is written as a result.
In general I find the paper well written, few details that should be corrected.
Reviewer 3 Report
The research titled 'Genetic diversity analysis and breeding of Local breeds based on the mitochondrial ND6 gene' has a very interesting topic. I found the article's subject matter to be fascinating, and I read the manuscript with great interest. The paper aligns well with the scope of the journal. The lack of line numbering makes it a bit difficult to review the article. Nevertheless, I have only one minor comment to make to the authors: I suggest replacing the first keyword 'Goose' with 'Local breeds.'
Author Response
请参阅附件。
